# Learning to Execute: Efficiently Learning Universal Plan-Conditioned Policies in Robotics

**Ingmar Schubert**[1], **Danny Driess**[1], **Ozgur S. Oguz**[2], and **Marc Toussaint**[1]

[1] Learning and Intelligent Systems Group, TU Berlin, Germany

[2] Machine Learning and Robotics Lab, University of Stuttgart, Germany

`{ingmar.schubert@,danny.driess@campus.,toussaint@}tu-berlin.de`

`ozgur.oguz@ipvs.uni-stuttgart.de`

## Abstract

Applications of Reinforcement Learning (RL) in robotics are often limited by high data demand. On the other hand, approximate models are readily available in many robotics scenarios, making model-based approaches like planning a data-efficient alternative. Still, the performance of these methods suffers if the model is imprecise or wrong. In this sense, the respective strengths and weaknesses of RL and model-based planners are complementary. In the present work, we investigate how both approaches can be integrated into one framework that combines their strengths. We introduce Learning to Execute (L2E), which leverages information contained in approximate plans to learn universal policies that are conditioned on plans. In our robotic manipulation experiments, L2E exhibits increased performance when compared to pure RL, pure planning, or baseline methods combining learning and planning.

## 1   Introduction

A central goal of robotics research is to design intelligent machines that can solve arbitrary and formerly unseen tasks while interacting with the physical world. Reinforcement Learning (RL) (Sutton & Barto, 2018) is a generic framework to automatically learn such intelligent behavior with little human engineering. Still, teaching an RL agent to actually exhibit general-purpose problem-solving behavior is, while possible in principle, prohibitive in practice. This is due to practical restrictions including limited computational resources and limited data availability. The latter limitation is particularly dramatic in robotics, where interaction with the physical world is costly.

On the other hand, for many robotics scenarios, there is a rough model of the environment available. This can be exploited, e.g., using model-based planning approaches (Mordatch et al., 2012; Kuindersma et al., 2016; Toussaint et al., 2018). Model-based planners potentially offer a more data-efficient way to reason about an agent's interaction with the world. Model-based planners have been used in many areas of robotics, such as for indoor and aerial robots (Faust et al., 2018), visual manipulation (Jeong et al., 2020), or humanoid walking (Mordatch et al., 2015). Still, if the model does not account for stochasticity or contains systematic errors, directly following the resulting plan will not be successful.

The present work starts from the observation that both pure RL approaches and pure planning approaches have strengths and limitations that are fairly complementary. RL makes no assumptions about the environment but is data-hungry, and model-based planning generally implies model simplifications but is data-efficient. For robotic manipulation tasks, it seems natural to try and integrate both approaches into one framework that combines the strengths of both. In the present work we seek to add an additional perspective to the open question of how this can be achieved best.

35th Conference on Neural Information Processing Systems (NeurIPS 2021).

We introduce a novel approach that we call Learning to Execute (L2E). Our approach translates sparse-reward *goal-conditioned* Markov Decision Processes (MDPs) (Bellman, 1957) into *plan-conditioned* MDPs. L2E exploits a simple planning module to create crude plans, which are then used to teach any off-the-shelf off-policy RL agent to execute them. L2E makes use of final-volume-preserving reward shaping (FV-RS) (Schubert et al., 2021), allowing it to train a universal plan-conditioned policy with high data efficiency. The contributions of this work are:

- We introduce L2E, which uses RL to efficiently learn to execute approximate plans from a model-based planner in a plan-conditioned MDP. We describe formally how FV-RS can be used as a tool to construct such plan-conditioned MDPs from goal-conditioned MDPs.
- We introduce plan replay strategies to efficiently learn universal plan-conditioned policies.
- We demonstrate, using robotic pushing problems, that L2E exhibits increased performance when compared to pure RL methods, pure planning methods, or other methods combining learning and planning.

We discuss work related to ours in section 2, explain background and notation in section 3, and introduce our method in section 4. We present our experimental results in section 5, discuss limitations in section 6, and conclude with section 7.

## 2  Related Work

### 2.1  Goal-Conditioned Policies

Goal-conditioned or universal policies (Kaelbling, 1993; Moore et al., 1999; Foster & Dayan, 2002; Schaul et al., 2015; Veeriah et al., 2018; Nasiriany et al., 2019) not only act based on the state the agent finds itself in, but also based on the goal it tries to achieve. Hindsight Experience Replay (HER) (Andrychowicz et al., 2017) is a particularly efficient way to learn universal policies. Here, achieved outcomes of the agent's interaction with the environment are interpreted as desired goals in order to improve sample efficiency in sparse-reward settings.

L2E draws great inspiration from this work, but in contrast to HER, L2E learns a universal plan-conditioned policy. This means that the L2E policy in general can execute multiple plans leading to the same goal. Although this presents a more complex learning task, we show in our experiments that by incorporating plan information using plan-based FV-RS, the sample efficiency of L2E is significantly improved over HER.

### 2.2  Plan- and Trajectory-Conditioned Policies

Plan-conditioned policies create behavior that depends on plans that are input to the decision making. Lynch et al. (2020) learn plans and how to execute them from data generated by a human "playing" with a teleoperated robot. The resulting policy is conditional on a latent space of encoded plans. Our work differs from this paradigm in that human interaction is not needed. Both Lynch et al. (2020) and Co-Reyes et al. (2018) directly imitate a planned trajectory by maximizing its likelihood. In contrast, the plans used in the present work are not directly imitated. Using FV-RS guarantees that the fully trained L2E agent will reach its goal after finite time even if the plan provided is wrong. Guo et al. (2019) learn trajectory-conditioned policies to self-imitate diverse (optimal and suboptimal) trajectories from the agent's past experience.

We instead assume in this work that the plan is provided by an external model-based planner. This allows the L2E agent to use external information during training that could not be concluded from its own experience yet.

### 2.3  Learning from Demonstration

L2E learns how to execute plans in order to achieve different tasks. In this sense, it is related to Learning from Demonstration (LfD) techniques that exploit demonstrations when learning a task. Existing work (Argall et al., 2009; Hussein et al., 2017; Ravichandar et al., 2020) differs significantly both in how the demonstration examples are collected and how the policy is then derived. Taylor et al. (2011) derive an approximate policy from human demonstration, and then use this to bias the

exploration in a final RL stage. Hester et al. (2017) train a policy on both expert data and collected data, combining supervised and temporal difference losses. Salimans & Chen (2018) use a single demonstration as starting points to which the RL agent is reset at the beginning of each episode. Peng et al. (2018) use motion capture data to guide exploration by rewarding the RL agent to imitate it. In Cabi et al. (2019), demonstrations are combined with reward sketching done by a human. Interactive human feedback during training is another source of information used in Thomaz et al. (2006); Knox & Stone (2010). Kinose & Taniguchi (2020) integrate RL and demonstrations using generative adversarial imitation learning by interpreting the discriminator loss as an additional optimality signal in multi-objective RL.

While these LfD approaches are related to L2E in that external information is used to increase RL efficiency, it is in contrast assumed in L2E that this external information is provided by a planner.

## 2.4 Combining Learning with Planning

Similarly to demonstrations, external plans can be exploited to facilitate learning. Faust et al. (2018) connect short-range goal-conditioned navigation policies into complex navigation tasks using probabilistic roadmaps. In contrast, L2E learns a single plan-conditioned policy for both short-term and long-term decision making. Sekar et al. (2020) use planning in a learned model to optimize for expected future novelty. In contrast, L2E encourages the agent to stay close to the planned behavior. Zhang et al. (2016) use model-predictive control to generate control policies that are then used to regularize the RL agent. In L2E, no such intermediate control policy is created, and a reward signal is computed directly from the plan. In Guided Policy Search (Levine & Koltun, 2013), differential dynamic programming is used to create informative guiding distributions from a transition model for policy search. These distributions are used to directly regularize the policy in a supervised fashion, while L2E makes use of FV-RS as a mechanism to interface planning and RL. Christiano et al. (2016) learn an inverse dynamics model to transfer knowledge from a policy in the source domain to a policy in the target domain. The idea of integrating model-based and model-free RL has also been studied independently of planning (Pong et al., 2018; Janner et al., 2019). In contrast, in L2E the model is translated by a planner into long-horizon plans.

In the experiments section, we compare L2E against two representative examples from the literature mentioned above. The first is using a plan to identify subgoals that are then pursued by an RL agent, as done in Faust et al. (2018). The second is executing the plan using an inverse model, similar to the approach in Christiano et al. (2016). These two baselines and L2E can be seen as representatives of a continuum: Christiano et al. (2016) follow the plan very closely, trying to imitate the planner at each time step. Faust et al. (2018) relax this requirement and only train the agent to reach intermediate goals. Finally, in L2E, the agent is free to deviate arbitrarily from the plan (although it is biased to stay close), as long as it reaches the goal. We find that L2E results in significantly higher success rates when compared against both baselines.

## 3 Background

### 3.1 Goal-Conditioned MDPs and RL

We consider settings that can be described as discrete-time MDPs $M = \langle \mathbb{S}, \mathbb{A}, T, \gamma, R, P_S \rangle$. $\mathbb{S}$ and $\mathbb{A}$ denote the set of all possible states and actions, respectively. $T : \mathbb{S} \times \mathbb{A} \times \mathbb{S} \to \mathbb{R}_0^+$ is the transition probability (density); $T(s'|s, a)$ is the probability of the next state being $s'$ if the current state is $s$ and $a$ is chosen as the action. The agent receives a real-valued reward $R(s, a, s')$ after each transition. Immediate and future rewards are traded off by the discount factor $\gamma \in [0, 1)$. $P_S : \mathbb{S} \to \mathbb{R}_0^+$ is the initial state distribution.

The goal of RL is to learn an optimal policy $\pi^* : \mathbb{S} \times \mathbb{A} \to \mathbb{R}_0^+$ that maximizes the expected discounted return. In other words, RL algorithms generally try to find

$$\pi^* = \underset{\pi}{\operatorname{argmax}} \sum_{t=0}^{\infty} \gamma^t \mathbb{E}_{s_{t+1} \sim T(\cdot|s_t, a_t),\, a_t \sim \pi(\cdot|s_t), s_0 \sim P_S} \left[ R(s_t, a_t, s_{t+1}) \right] \qquad (1)$$

from collected transition and reward data $D = \{(s_i, a_i, r_i, s_i')\}_{i=0}^n$. More specifically for this work, we are interested in applications in robotics, where both $\mathbb{S}$ and $\mathbb{A}$ are typically continuous. There

exists a wide range of algorithms for this case. For the experiments in this paper, soft actor-critic (SAC) (Haarnoja et al., 2018) is used.

In a goal-conditioned MDP $M_G = \langle \mathbb{S}, \mathbb{G}, \mathbb{A}, T, \gamma, R_G, P_S, P_G \rangle$, the reward function $R_G(s, a, s', g)$ has an additional input parameter, the goal $g \in \mathbb{G}$. Here, $P_G : \mathbb{G} \to \mathbb{R}_0^+$ is the distribution of goals. The optimal goal-conditioned policy $\pi_G^*$ acts optimally with respect to any of these goals.

### 3.2 Final-Volume-Preserving Reward Shaping

We use approximate plans as an additional source of information for the RL agent. For sparse-reward goal-driven MDPs, FV-RS (Schubert et al., 2021) offers an efficient way to include additional information by adding an additional term

$$R(s, a, s') \to R_{\text{FV}}(s, a, s') = R(s, a, s') + F_{\text{FV}}(s, a, s') \tag{2}$$

to the reward function, accelerating exploration. In general, the optimal policy $\pi^*$ corresponding to the original MDP and the optimal policy $\pi_{\text{FV}}^*$ corresponding to the shaped MDP will be different. FV-RS however restricts the allowed modifications $F_{\text{FV}}(s, a, s')$ in such a way that after finite time, the optimally controlled agent ends up in a subset of the volume in which it would have ended up without shaping. As a result, external information can be made available for the RL algorithm without changing the long-term behavior of the resulting optimal policy.

Specifically in the present work, we consider goal-conditioned MDPs in which the goal-conditioned reward $R_G$ of the underlying MDP is either $1$, if the goal is reached, or $0$ everywhere else. We further assume that the L2E agent is given an external plan $p$, represented as an intended trajectory $p = (p_1, p_2, \dots)$ in state space. We intend to reward the agent for staying close to the plan, and for advancing towards the goal along the plan. A natural way of achieving this is to use a plan-based shaping reward (Schubert et al., 2021). The single-plan shaping function introduced there can be generalized to the multi-plan setting in the present work in the following way:

$$F_{\text{FV}}(s, a, s', p) = \frac{1 - R_G(s, a, s', f(p))}{2} \frac{k(s) + 1}{L} \exp\left(-\frac{d^2(s, p_{k(s)})}{2\sigma^2}\right) \tag{3}$$

Here, $f(p)$ denotes the goal that $p$ leads to, $\sigma \in (0, \infty)$, $k(s) = \operatorname{argmin}_i(d(p_i, s))$, and $d(\cdot, \cdot)$ is a measure of distance in state space. For the pushing experiments discussed in this work, $d(\cdot, \cdot)$ is the euclidean distance in state space ignoring the coordinates corresponding to the orientation of the box. The first term in eq. (3) ensures that the assigned shaping reward $F_{\text{FV}}$ is always smaller than the maximum environment reward (at most $1/2$), and that if the binary environment reward is $1$, no shaping reward is assigned. The second term rewards the agent for advancing towards the goal along the plan, and the third term rewards the agent for staying close to the plan. For a sufficiently high discount factor $\gamma$, $F_{\text{FV}}$ is final-volume preserving, meaning that the long-term behavior of the optimal agent is unchanged.

## 4 Learning to Execute

L2E considers goal-conditioned MDPs $M_G$ (see section 3.1), for which an approximate planner $\Omega$ is available. L2E uses FV-RS to construct a corresponding plan-conditioned MDP $M_P$ from a goal-conditioned MDP $M_G$ and a planner $\Omega$. In the following sections 4.1 to 4.3, we introduce our notion of a plan-conditioned MDP $M_P$ and describe the components of the L2E algorithm. We then summarize the L2E algorithm in section 4.4.

### 4.1 Plan-Conditioned MDPs

Plans are provided by a model-based planner, which can be described as a distribution $\Omega : \mathbb{P} \times \mathbb{S} \times \mathbb{G} \to \mathbb{R}_0^+$ over a set of plans $\mathbb{P}$. Given an initial state and a goal, $\Omega(p|s, g)$ is the probability that the planner outputs $p$ as a possible plan of how to achieve $g$ from state $s$. The distinction between goals and plans is that plans are conditional on *both a goal and an initial state*. Therefore, both initial state and goal can be inferred using the plan only.

In a plan-conditioned MDP $M_P = \langle \mathbb{S}, \mathbb{P}, \mathbb{A}, T, \gamma, R_P, P_S, P_P \rangle$, a plan $p \in \mathbb{P}$ is given to the reward function $R_P(s, a, s', p)$ as an additional input parameter. $P_P : \mathbb{P} \to \mathbb{R}_0^+$ is the distribution of plans. The optimal plan-conditioned policy $\pi_P^*$ behaves optimally with respect to any of these plans, creating a distribution $\pi_P^*(\cdot \mid s, p)$ over actions that is conditional on the current state and the current plan.

## 4.2 Constructing the Plan-Conditioned MDP

We use FV-RS to shape the reward function $R_G$ of the original goal-conditioned MDP $M_G = \langle \mathbb{S}, \mathbb{G}, \mathbb{A}, T, \gamma, R_G, P_S, P_G \rangle$ with a plan-dependent term $F_{\text{FV}}(s, a, s', p)$ (see equation 3)

$$R_G(s, a, s', g) \to R_G^{\text{FV}}(s, a, s', g, p) = R_G(s, a, s', g) + F_{\text{FV}}(s, a, s', p) \quad . \tag{4}$$

We call $g = f(p)$ the goal for which the plan p was created. If a planner $\Omega$ should be such that g can not be recovered from the resulting plan $p \sim \Omega(.|s, g)$, we can always construct a new $\tilde{p} \sim \tilde{\Omega}$ such that $\tilde{p} = [p, g]$. Since now g can be recovered from $\tilde{p}$ deterministically, we can assume that $f$ always exists without loss of generality. We can interpret the shaped reward function

$$R_P(s, a, s', p) = R_G^{\text{FV}}(s, a, s', f(p), p) \tag{5}$$

as a plan-conditioned reward function of a plan-conditioned MDP $M_P = \langle \mathbb{S}, \mathbb{G}, \mathbb{A}, T, \gamma, R_P, P_P \rangle$. The distribution over initial states and plans $P_P$ of $M_P$ is still missing, and can be constructed as

$$P_P(s, p) = \int \Omega(p|s, g) P_S(s) P_G(g) \mathrm{d}g \quad . \tag{6}$$

In practice, $P_P$ can be sampled from by first sampling $s \sim P_S$, $g \sim P_G$ and then subsequently sampling $p \sim \Omega(\cdot|s, g)$.

Thus, we have constructed a plan-conditioned MDP $M_P$ by combining a goal-conditioned MDP $M_G$ with an approximate planner $\Omega$ and a FV-RS shaping function $F_{\text{FV}}$. For reference later in this paper, we write as a shorthand notation $M_P = \mathcal{C}(M_G, \Omega, F_{\text{FV}})$. Furthermore, we will refer to $M_P$ as the corresponding plan-conditioned MDP to $M_G$ and vice versa.

In contrast to potential-based reward shaping (Ng et al., 1999), FV-RS does not leave the optimal policy invariant. As a result, generally $\exists p \in \mathbb{P} : \pi_G^*(\cdot|\cdot, f(p)) \not\equiv \pi_P^*(\cdot|\cdot, p)$. In words, the optimal policy of $M_P$ and the optimal policy of $M_G$ will not result in identical behavior. In fact, while $\pi_G^*(\cdot|\cdot, g)$ learns one policy for each goal $g$, $\pi_P^*(\cdot|\cdot, p)$ can learn different behavior for each plan in the set of plans $\{p \in \mathbb{P} \mid f(p) = g\}$ leading towards the same goal $g$.

## 4.3 Plan Replay Strategy

In order to efficiently learn a universal plan-conditioned L2E policy, the reward for experienced episodes is evaluated with respect to many different plans. In HER (Andrychowicz et al., 2017), it is assumed that each state $s \in \mathbb{S}$ can be assigned an achieved goal. Recorded episodes are then replayed with respect to goals that were achieved during the episode, i.e. the recorded transitions are re-evaluated with respect to these goals. This ensures that the recorded transitions were successful in reaching the replayed goals, resulting in highly informative data.

In L2E, transitions are replayed with respect to plans. However, there is no meaningful relation between each state $s \in \mathbb{S}$ and a unique "achieved plan". Therefore, the L2E agent replays transitions with past plans that were recorded at some point during training and were stored in its replay buffer $D$. The replay plans are chosen according to a plan replay strategy $S_n$.

A plan replay strategy $S_n$ provides a distribution over $n$ replay plans, conditioned on the replay buffer $D$ and the buffer containing the current episode $D_{\text{ep}}$ (see algorithm 1 for a definition of $D$ and $D_{\text{ep}}$). For replay, $n$ plans are sampled according to this strategy $\{p_1, \ldots, p_n\} \sim S_n(\cdot \mid D_{\text{ep}}, D)$. We consider two types of replay strategies. Uniform replay $S_n^{\text{uni}}$ samples $n$ unique plans uniformly from the replay buffer $D$. Reward-biased replay $S_{nm}^{\text{bias}}$ first uniformly samples $m$ unique plans from the replay buffer $D$, and then returns the $n$ plans $p_i$ that would have resulted in the highest sum of rewards $\sum_{(s_k, a_k, s_k') \in D_{\text{ep}}} R_P(s_k, a_k, s_k', p_i)$ for the episode stored in $D_{\text{ep}}$. The idea behind using reward-biased replay is to bias the replay towards transitions resulting in higher reward.

## 4.4 L2E Algorithm

The L2E algorithm is outlined in algorithm 1. First, the corresponding plan-conditioned MDP $M_P = \mathcal{C}(M_G, \Omega, F_{\text{FV}})$ is constructed from the original goal-conditioned MDP $M_G$, the planner $\Omega$ and the shaping function $F_{\text{FV}}$ as described in section 4.2. The agent acts in the environment trying to follow one randomly sampled plan per episode. The episode is then added to the replay buffer,

---

**Algorithm 1:** Learning to Execute (L2E)

---

**Input** : Goal-conditioned MDP $M_G$, approximate planner $\Omega$, FV-RS shaping function $F_{\text{FV}}$,
plan replay strategy $S_n$, off-policy RL Algorithm $\mathcal{A}$

**Output** : Universal plan-conditioned optimal policy $\pi_P^*$ for the corresponding plan-conditioned
MDP $M_P = \mathcal{C}(M_G, \Omega, F_{\text{FV}})$

---

**1** Construct plan-conditioned MDP $M_P = \mathcal{C}(M_G, \Omega, F_{\text{FV}})$ as detailed in section 4.2;
**2** Initialize replay buffer $D \leftarrow \{\}$;
**3** **while** $\pi_P^*$ *not converged* **do**
**4**     Initialize episode buffer $D_{\text{ep}} \leftarrow \{\}$;
**5**     Sample initial state and goal $(s_0, g) \sim P_G$;
**6**     Sample plan $p \sim \Omega(\cdot | s_0, g)$;
**7**     $s \leftarrow s_0$;
**8**     **while** *Episode not done* **do**
**9**        Sample action $a \sim \pi_P^*(\cdot \mid s, p)$;
**10**       Sample transition $s' \sim T(\cdot \mid s, a)$;
**11**       Collect shaped reward $r \leftarrow R_P(s, a, s', p)$;
**12**       Add to episode buffer $D_{\text{ep}} \leftarrow D_{\text{ep}} \cup \{(s, a, r, s', p)\}$;
**13**       $s \leftarrow s'$;
**14**     **end**
**15**     Add episode to replay buffer $D \leftarrow D \cup D_{\text{ep}}$;
**16**     Get replay plans $\{p_1, \ldots, p_n\} \sim S_n(\cdot \mid D_{\text{ep}}, D)$;
**17**     **for** $p_{replay}$ *in* $p_1, \ldots, p_n$ **do**
**18**       **for** $(s, a, r, s', p)$ *in* $D_{ep}$ **do**
**19**         Calculate replay reward $r_{\text{replay}} \leftarrow R_P(s, a, s', p_{\text{replay}})$;
**20**         Add replayed transition to buffer $D \leftarrow D \cup \{(s, a, r_{\text{replay}}, s', p_{\text{replay}})\}$;
**21**       **end**
**22**     **end**
**23**     Update policy using off-policy RL algorithm $\pi_P^* \leftarrow \mathcal{A}(\pi_P^*, D)$
**24** **end**

---

along with data from episode replays with respect to other plans. These other plans are sampled from the replay buffer according to the replay strategy $S_n$. A generic off-policy RL algorithm is used to update the agent using the replay buffer. This process is repeated until convergence.

We would like to emphasize that the L2E algorithm is agnostic to the exact type of off-policy RL algorithm. By combining state and plan into a "super state" for the purpose of passing the replay buffer to the off-policy RL algorithm, L2E can be interfaced with any off-the-shelf implementation.

## 5 Experiments

We evaluate the L2E agent against several baselines using two simulated robotic manipulation tasks, namely a pushing task and an obstacle avoidance task. These two environments are chosen to compare different approaches on a variety of challenges. While the pushing task can be seen as an open-source version of the opanAI gym FetchPush-v1 task (Brockman et al., 2016), the obstacle task was chosen to represent robotic manipulation tasks with segmented state spaces. This allows us to discuss limitations of exploration in such environments as well.

A video of the experiments is available in the supplementary material. The complete code to fully reproduce the figures in this paper from scratch can be found at github.com/ischubert/l2e and in the supplementary material. This includes the implementation of the environments, the implementation of the L2E agents and the baselines, and the specific code used for the experiments in this paper.

The experiments section is structured as follows. In section 5.1 we discuss the environments and planners that are used in the experiments. We briefly introduce the plan embedding used for the L2E agent in section 5.2, additional experiments on this can be found in section A.5 In section 5.3 we introduce the baselines against which we compare our method. In section 5.4 we discuss our experimental results. Implementation details of the L2E agent are given in section A.1

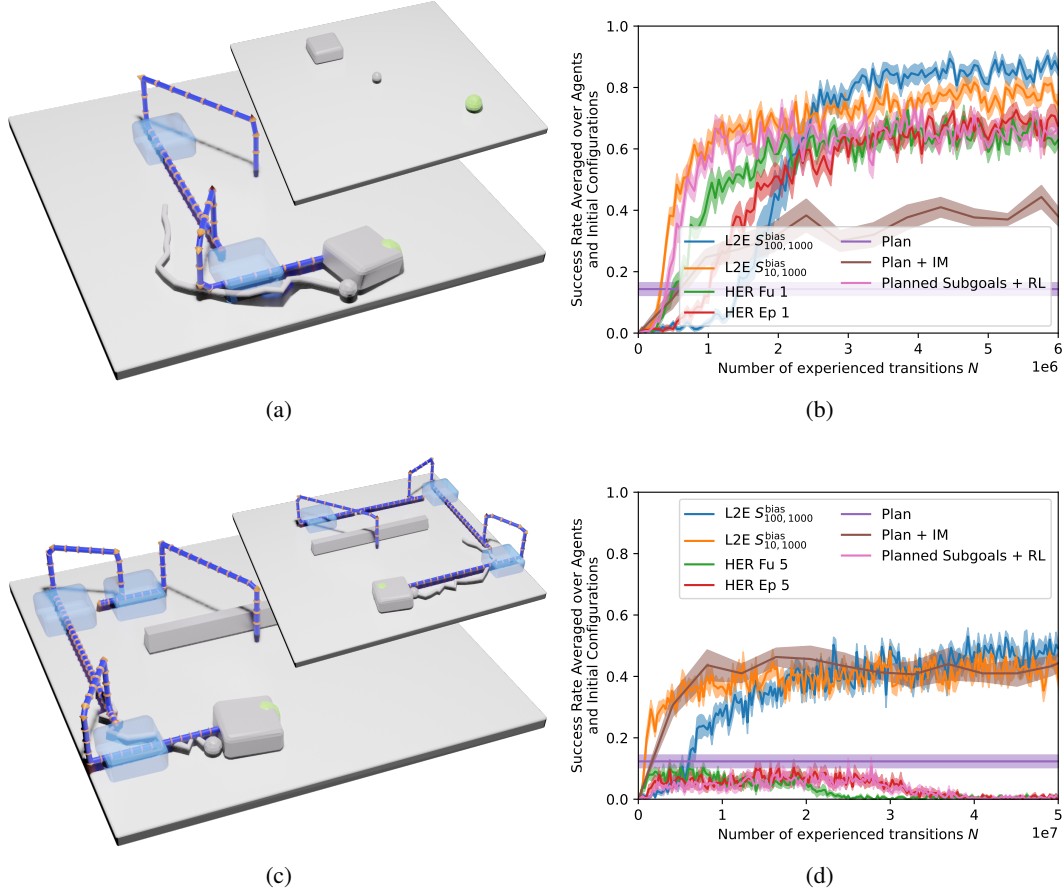

(a)

(b)

(c)

(d)

Figure 1: Results for the basic pushing environment (figures (a) and (b)) and the obstacle pushing environment (figures(c) and (d)). Lines represent averages over $N$ independently trained agents times $M$ test rollouts per agent and time step, the colored areas indicate the confidence intervals (standard deviations of the mean). Renderings created with open-source code by Orthey (2021). (a) The simulated pushing setup (inset). The spherical end effector tries to push the box to the green target. Once the box is pushed off the table, it can not be recovered. An approximate planner creates simple manhattan-like plans (blue line and transparent boxes in large figure; the end effector is planned to change height to push the box from the other side). The path of the box is planned fully, but only shown at some time steps. Recent end effector movement is indicated in grey. Learned behavior can differ significantly from the original plan. (b) Comparison to the baselines introduced in section 5.3. L2E shows significantly higher sample efficiency and success rates. See section A.7 for a longer run of L2E $S^{\text{bias}}_{100,1000}$. (c) In the simulated obstacle setup, an additional obstacle is added. In this environment, the planner can create (and the RL agent learns to execute) multiple plans from the same initial configuration to the same goal (inset shows alternative plan). (d) Comparison to the baselines introduced in section 5.3. L2E performs significantly better than the pure learning HER baselines, the pure planning baseline ("Plan"), and the "Planned Subgoals + RL" baseline. While using an inverse model is initially more efficient, L2E achieves significantly better results if given enough data.

## 5.1 Environments and Planners

Figure 1a and Figure 1c show renderings of the basic pushing environment and obstacle pushing environment, respectively. We use the open-source Nvidia PhysX engine (phy, 2021) to simulate a box of size $0.4 \times 0.4$ being pushed on a table of size $3 \times 3$ by a spherical end effector of radius $0.06$. The 10D state space of both the goal-conditioned MDP $M_G$ and the corresponding plan-conditioned MDP $M_P$ consists of the 3D position of the end effector, the 3D position of the box, and the 4D quaternion for the orientation of the box. The agent controls the 3D velocity of the end effector. The

maximum velocity in any direction is $0.1$ per time step. The end effector movement resulting from the agent's actions is slightly distorted by random noise. In the obstacle pushing environment, the agent additionally has to evade an obstacle in the middle of the table.

In the goal-conditioned MDP $M_G$, each goal is represented as a desired 2D box position on the table. The goal-dependent sparse reward function $R_G$ is 1 if the box is within $0.1$ of this desired goal, and 0 if not. The initial state-goal distribution $P_G$ is uniform across the table for the non-colliding box position and goal position. The end effector is always initialized at the origin and the box is always initialized with a fixed orientation parallel to the table.

For the basic pushing environment, we use a crude manhattan-like planner $\Omega$ that deterministically outputs plans consisting of two separate contacts leading the initial state to the goal as shown in Figure 1a. For the obstacle pushing environment, plans consist of four contacts, corresponding to an additional intermediate box position which is chosen at random (see Figure 1c). Thus, the agent learns to execute an infinite number of plans for each combination of start and goal.

Plans are represented as a trajectory of length 50 for the basic pushing environment and 100 for the obstacle pushing environment, consisting of 6D elements representing end effector position and box position. For the basic pushing environment, we additionally report results for less dense plans in section A.6. The orientation of the box is not specified in the plans. We construct the plan-conditioned MDP $M_P$ as described in section 4.2, using this planner and the FV-RS function in equation 3. We use the width parameter $\sigma = 0.5$ throughout the experiments.

## 5.2 Plan Encoding

The plans $p$ are embedded before they are provided to the policy. A plan encoding is an injective function $\phi : \mathbb{P} \to \mathbb{C}$ from the set of plans $\mathbb{P}$ to a latent space $\mathbb{C}$. If $\mathbb{P}$ is a manifold in some high-dimensional space, the dimensionality of the latent space must be at least as high as the dimensionality of the manifold. Since $\mathbb{P}$ is task-dependent, the encoding will be task-dependent as well. For the basic pushing environment (Figure 1a), $\mathbb{P}$ is a 4D manifold (since the plans only depend on the initial and final 2D box positions). For the obstacle task (Figure 1c), $\mathbb{P}$ is a 6D-manifold (since the plans depend on one intermediate box position as well).

In the experiments discussed in the present work, we encode plans analytically using box positions as described above. We experimentally compare this with either learning the encoding or not using any encoding at all in section A.5.

## 5.3 Baselines

We compare L2E against (1) direct plan execution, (2) plan execution with an inverse dynamics model, (3) using RL to reach subgoals, and (4) HER. We describe these baselines in detail in section A.2.

## 5.4 Results

Both at training and evaluation time, we run episodes of length 250. For each method $q$ (i.e., L2E and all baselines), we independently train $A = 10$ agents. After $N$ environment transitions, we evaluate the agents. We reset to random initial positions and goals/plans and run the experiment until the goal is reached or until the episode ends. We repeat this process $M = 30$ times for each agent, and store whether the rollout was successful in reaching the goal. We denote the result of the $m$-th evaluation of the $a$-th agent for method $q$, evaluated after learning for $N$ environment transitions, as $\mathcal{F}_{am}^{(q)}(N)$.

As can be seen from the video given in the supplementary material, even though the L2E agent uses plan information as a guiding bias during exploration, and is encouraged to stay close to the plan by the shaping reward, it can also learn to deviate considerably from the plan if closely following it will be suboptimal for reaching the goal fast. For example, while the simple planner (see Figure 1a and Figure 1c) suggests to re-establish the contact during the sequence, the L2E agent almost always moves and turns the box using a single contact.

### 5.4.1 Basic Pushing Environment

To allow for a fair comparison, we spent a considerable amount of effort to optimize the HER replay strategy as well as the L2E strategy. Details on this are given in section A.4.

The results for the pushing setup are summarized in Figure 1b. We observe that both L2E versions outperform all baselines in terms of the asymptotical performance. L2E with biased replay strategy $S_{10,1000}$ exhibits a high sample efficiency especially in the beginning, resulting in success rates significantly higher than $50\%$ after $4000$ episode rollouts or $1$ Million time steps. Directly executing the plan results in very low success rates of significantly less than $20\%$ on average. Executing the plan with an inverse model (IM) still shows significantly worse long-term performance than the RL methods. HER results in better policies than the IM baselines, but is relatively data hungry. This can be improved slightly if the HER agent is only used to reach subgoals given by the planner.

Pushing is a challenging interaction that requires reasoning for several time steps ahead. A typical failure mode of the IM baseline (see also videos) is that the box moves away from the intended trajectory too much, so that the agent is not able to correct for it within one time step. In contrast, the L2E agent learns to deviate from the planned trajectory if this is required to reach the goal.

We find that L2E, combining a model-based planner and a universal plan-conditioned policy, outperforms our baselines that are pure planning or pure learning approaches. In addition, L2E outperforms the two baselines that also combine learning and planning.

### 5.4.2 Obstacle Pushing Environment

L2E performs significantly better than the pure learning HER baselines, the pure planning baseline ("Plan"), and the "Planned Subgoals + RL" baseline. While using an inverse model is initially more efficient, L2E achieves significantly better results if given enough data.

Comparing the basic pushing environment (section 5.4.1) to the obstacle environment, L2E learns slower in the latter. This is in part due to the higher dimensionality of the latent space of plan encodings (see also section 5.2), posing a more challenging learning problem to the L2E agent. In contrast, the "Plan+IM" baseline is independent of the size of the plan space, and performs comparably to the experimental setting in the original version.

The obstacle in the middle segments the state space into two parts. In order to move from one side to the other, an agent already has to be able to reason about long-term results of its actions. As evidenced by the results for HER, this poses a significant challenge for pure RL. Incorporating planner knowledge helps the agent to overcome this chicken-and-egg problem.

## 6 Discussion

Learning plan-dependent policies as opposed to goal-dependent policies has the additional advantage that the former can learn to execute multiple plans that lead from the same initial state to the same goal, as shown in the obstacle environment. Thus, the policy learns multiple strategies to achieve the same outcome. In principle, this allows it to adapt to changed scenarios where some of these strategies become infeasible. If, e.g., the environment changes, it suffices to only update the planner's crude model of the environment so that it creates plans that are feasible again. These can then be directly fed into the policy without retraining. We explore this possibility in section A.3, using a simple 2D maze environment with moving obstacles. We find that the plan-conditioned L2E policy consistently achieves 90% success rate in this quickly changing environment, while the goal-conditioned HER policy does not improve beyond 60% success rate.

We used rather simple plans to support the RL agent during training, and demonstrated that these are already sufficient to significantly speed up learning in our experiments. In fact we demonstrate in section A.6 that in the basic pushing example, the L2E agent is very robust against plans of even lower quality. Using simple plans enabled us to use an analytical encoding; for very complex scenarios it might be beneficial to learn the encoding using an auxiliary objective (see, e.g., Co-Reyes et al. (2018)). We present results on using a variational autoencoder (VAE) in section A.5.

The use of FV-RS biases the RL agent towards following the plan. While it was shown in the experiments that the RL agent can learn to deviate from the plan, plans that are globally misleading can act as a distraction to the agent. In the present work, it is assumed that plans can be used to guide the agent during learning, increasing sample efficiency. Independently of the specific method used to achieve this, misleading plans will always break this assumption.

Comparing the basic pushing environment to the obstacle pushing environment, the amount of data needed for learning a plan-conditioned policy clearly depends on the size of the plan spaces that are considered. For very large plan spaces, more data will be needed to master the task. Still, including planner information into the learning process makes a decisive difference, as demonstrated by the relative performance of L2E and HER in the obstacle example.

While SAC was used for the experiments in section 5, L2E can be used in combination with any off-policy RL algorithm. L2E reformulates a goal-conditioned MDP as a plan-conditioned MDP, and provides a replay strategy to efficiently solve the latter. It is agnostic to how this data is then used by the RL agent.

The specific FV-RS shaping function used in this work applies to MDPs with sparse rewards. We focused on this since sparse rewards are common in robotic manipulation. In addition, they often present notoriously hard exploration tasks, making external plan-based information as used by L2E particularly useful. However, FV-RS in general is not restricted to sparse-reward settings, and by using a different shaping function, L2E could be applied in other settings as well.

Apart from FV-RS, there are alternative schemes of reward shaping such as potential-based reward shaping (PB-RS) Ng et al. (1999). In principle, these could also be used to increase the sample efficiency of the RL agent. We chose FV-RS for two reasons. First, in the original paper Schubert et al. (2021), it was demonstrated that FV-RS leads to significantly higher sample efficiency than PB-RS. Second, since PB-RS leaves the optimal policy invariant, the behavior of the fully converged policy trained with PB-RS will only be goal-dependent, and not depend on the rest of the plan.

The original HER paper (Andrychowicz et al., 2017) considers the use of a simple form of reward shaping in combination with HER as well. It is found that reward shaping dramatically reduces the performance of HER in a robotic pushing task. In the present work, we show in contrast that including plan information using FV-RS shaping improves the performance of RL in a similar task. A possible explanation to reconciliate these seemingly contradictory results is already offered by Andrychowicz et al. (2017): While simple domain-agnostic shaping functions can be a distraction for the RL agent, domain-specific reward shaping functions can be beneficial. This view is supported, e.g., by similar results by Popov et al. (2017). Andrychowicz et al. (2017) state that however "designing such shaped rewards requires a lot of domain knowledge". In this context, one could view L2E as an automated way to extract such domain-specific knowledge from model-based planners and make it available. We specifically believe that L2E can be useful in robotic manipulation tasks, where domain knowledge is in fact readily available in many cases. Here, L2E offers a way to exploit this.

## 7 Conclusion

We introduced L2E, an algorithm that links RL and model-based planning using FV-RS. RL generally results in well-performing policies but needs large amounts of data, while model-based planning is data-efficient but does not always result in successful policies. By combining the two, L2E seeks to exploit the strengths of both approaches. We demonstrated that L2E in fact shows both higher sample efficiency when compared to purely model-free RL, and higher success rates when compared to executing plans of a model-based planner. In addition, L2E also outperformed baseline approaches that combine learning and planning in our experiments.

## Acknowledgments and Disclosure of Funding

The authors would like to thank Valentin N Hartmann for stimulating discussions. The research has been supported by the International Max-Planck Research School for Intelligent Systems (IMPRS-IS), and by the German Research Foundation (DFG) under Germany's Excellence Strategy EXC 2120/1–390831618 "IntCDC" and EXC 2002/1–390523135 "Science of Intelligence".

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
