# OpenReview forum: "Learning to Execute: Efficient Learning of Universal Plan-Conditioned Policies in Robotics"
_NeurIPS.cc/2021/Conference — NeurIPS 2021 Poster_

### Official Review · Reviewer_CAJo · 2021-06-26

**Rating:** 7
**Confidence:** 4

**Summary:**

The paper proposes to condition policies on plans. Therefore, the policy learns to execute an approximate plan and correct it if necessary. This idea is instantiated in an algorithm that uses SAC for policy optimization and a "crude manhattan-like planner" for planning. The algorithm is evaluated on a box pushing task, where an improvement in sample-efficiency over goal-condition RL is shown, and an improvement in success rate compared to model-based plan execution with learned inverse dynamics model is demonstrated.

**Limitations And Societal Impact:**

-

**Main Review:**

The idea of conditioning policies on expert plans as presented in this paper is novel according to my knowledge. The paper is of good quality and it is written clearly. Together with the code in the supplementary materials, the paper provides sufficient details to reproduce the results. However, the evaluation is performed only in one environment and there is no discussion or evaluation of the influence of the function that encodes the plans before passing them to the policy. This lack of evaluations makes it hard to judge the significance of the proposed method as it is not clear if it generalizes to other settings. Since the main competitive approach considered in the paper is HER, it might make sense to add evaluations of the proposed L2E algorithm in the environments from the original HER paper.

Major comments
- line 231: "we analytically encode the plans into a 4D latent space". Provide the details of this mapping in the paper since this is one of the crucial elements of the proposed approach. Why 4D? What properties should this mapping satisfy? Does the dimensionality of the latent space depend on the task? Either theoretical study or empirical evaluations of the choices of this mapping should definitely be provided, because conditioning the policy on plans instead of goals is the main difference of the proposed method  compared to goal-conditioned RL.
- lines  272-276 say that L2E agent significantly deviates from the plans. This appears to imply that conditioning on the plan may not be so informative to the agent. What is the advantage of having a plan-conditioned policy instead of a goal-conditioned one? It is shown in the pushing experiment that with the chosen hyperparameters one achieves faster convergence, but are there any benefits in general that one should expect from feeding in plans instead of goals?
- it would be beneficial to the reader to contrast the proposed approach to guided policy search (GPS) and DeepMimic, as those methods also involve imitation of an imperfect teacher, although they don't explicitly feed the plans as input to the policy

Minor comments
- line 178: plan replay strategy S_n is defined as "a distribution over n replay plans". However, in lines 181-182 it is said that the uniform replay strategy samples n unique plans uniformly from the replay buffer D, which contains more than n plans. Therefore, I think the authors mean that S_n is a distribution over integers 1, 2, ..., len(D). So, the subscript n seems misleading in S_n because the distribution itself does not depend on how many samples are drawn from it.
- lines 198-199: what is meant by "unstable dynamics" in the pushing task? Every state of the system seems to be a stable equilibrium state if no control input is applied. Maybe a different term instead of "unstable" would be more appropriate here. The same in line 285.
- typo:  line 146, policy should be conditioned on state instead of action

Comments after rebuttal
I thank the authors for addressing my questions and  for providing additional experiments. I raise my score from 5 to 6. There are still quite a few concerns raised by other reviewers, such as providing comparisons to more directly related learning+planning algorithms instead of HER. Furthermore, the environment used for additional experiments is still a custom made environment, therefore it is hard to directly relate it to other papers. I am also not convinced by the argument that having a plan-conditioned policy is necessary to have a multi-modal policy: in soft Q-learning this is achieved by using an energy-based model for the policy, and if it is goal-conditioned, it may produce different solutions as samples from a multimodal distribution. For all these reasons, I can only say that the paper is marginally above the acceptance threshold.

Final comments
The authors addressed all my concerns and promised to add the results provided in the anonymous github repository to the paper. With these additions, I consider the paper appropriate for publication and raise my score to 7.

**Time Spent Reviewing:**

3h

---

> ### Author Response · Authors · 2021-08-10
> **Thank you for your review!**
>
> ## Q1: More environments
> We agree that more experiments would support the paper. We therefore added an additional task in the revised version. The pushing environment in the original version of the paper is an example with an “unsegmented” state space; in that respect it is similar to the other two examples considered in the original HER paper. Therefore, while we agree that more environments should be provided, we believe that the following choice might allow for a more diverse evaluation:
>
> In order to also test environments with a "segmented" state space, we now added a scenario where (1) an obstacle needs to be evaded and (2) stochasticity is increased significantly. Obstacle avoidance is a notoriously hard problem in robotics. The obstacle "segments" the state space, posing a significant challenge to exploration. We find that L2E shows improved performance compared to the HER baselines and the pure planning baseline. While currently the L2E runs are not converged yet, preliminary results show that the performance of L2E is already comparable to the performance of planning with a learned inverse model.
>
> The experiments are described and preliminary results are presented in an anonymized way here: https://github.com/paper6317authors/paper6317 Due to restricted space, we added the experimental details for these new environments in the appendix, and reference them in the experimental section. We updated the discussion (section 6) to now discuss the new experiments as well.
>
> ## Q2: Plan encoding
>
> We agree that this is worth discussing in detail. We added additional experiments, comparing analytical encoding as used originally to (1) using no encoding and (2) learning the encoding with a variational autoencoder (VAE) using a mean squared error reconstruction loss on the plan. See below for full results. We find that, while the analytical embedding performs significantly better than learning it using a VAE, a learned embedding is still advantageous compared to using no embedding at all (giving full plans to the policy).
>
> The experiments are described and preliminary results are presented in an anonymized way here: https://github.com/paper6317authors/paper6317 Due to restricted space, we added the experimental details for these new experiments in the appendix, and reference them in the experimental section. We slightly changed the discussion to now discuss these experiments as well.
>
> ## Q3: Plan encoding dimensionality
> We agree that this deserves more detail. We added a subsection before section 5.2 to describe the encoding used in the experiments in detail (please see below; text in square brackets is omitted in the paper). In this text, we also refer to additional experiments on other types of encoding, which have been added to the revised version (see Q2 for details).
>
> We added the following text before section 5.2:
>
> “5.2. Plan Encoding
>
> The plans p are embedded before they are provided to the policy. A plan encoding is an injective function \phi: \mathbb{P} \rightarrow \mathbb{C} from the set of plans \mathbb{P} to a latent space \mathbb{C}. If \mathbb{P} is a manifold in some high-dimensional space, the dimensionality of the latent space must therefore be at least as high as the dimensionality of the manifold. Since \mathbb{P} is task-dependent, the encoding will be task-dependent as well. For the pushing environment without obstacle [included in the original version], the plans are on a 4D manifold (since they only depend on the initial and final 2D box positions). For the obstacle task [added in the revised version], plans are on a 6D-manifold (since they depend on one intermediate box position as well).
>
> In the experiments discussed in the present work, we encode plans analytically using box positions as described above. We present experimental comparisons on either learning the encoding or not using any encoding at all in section A.3.“
>
> ## Q4: Advantage of having a plan-conditioned policy
> Thank you for pointing this out. We would like to answer to this in two parts:
>
> First a clarification of lines 272-276: While the L2E agent can, if needed, deviate from the plan significantly, the optimal behavior for the agent is to stay close to the plan where possible. In lines 272-276, our aim was to emphasize that the agent is not “forced” to stay close to the plan if such a behavior is not beneficial for reaching the goal fast. We agree that this was unclear, and we therefore updated lines 272-274 in the following way:
>
> “As can be seen from the video given in the supplementary, even though the L2E agent uses plan information as a guiding bias during exploration, and is encouraged to stay close to the plan by the shaping reward, it can also learn to deviate considerably from the plan if closely following it will be suboptimal for reaching the goal fast.”
>
> Second, to directly answer your question: An important benefit of feeding in plans instead of goals is that a policy can learn to follow multiple plans that all lead from the same start to the same goal (as demonstrated in the obstacle environment we added in the revised version, see https://github.com/paper6317authors/paper6317). This results in more flexibility, as multiple strategies can be tried out. We added the following text to the discussion:
>
> “As shown in the obstacle environment, an important difference between goal-conditioned policies and plan-conditioned policies is that the latter can learn to execute multiple plans that lead from the same initial state to the same goal. Thus, the policy learns multiple strategies to achieve the same outcome. In principle, this could allow it to adapt to changed scenarios where some of these strategies become infeasible.”
>
> ## Q5: Related work: Imitation of imperfect teacher
>
> We agree; we added a contrastive discussion of the mentioned work in the related work section of the revised version (see below). We split up section 2.3 into “2.3 Learning from Demonstration” and “2.4. Combining Learning and Planning”. In the revised version, we discuss DeepMimic in 2.3. in the following way:
>
> “In DeepMimic [Peng et al., 2018], motion capture data is used to guide exploration by rewarding the RL agent to imitate it. This additional reward signal bears some similarity to the present work, however, in L2E the reward signal is provided by an approximate planning module.“
>
> And we discuss GPS in section 2.4. in the following way:
>
> “In Guided Policy Search [Levine & Koltun 2013], differential dynamic programming is used to create informative guiding distributions for policy search from a transition model. These distributions are used to directly regularize the policy in a supervised fashion, while L2E makes use of FV-RS as a mechanism to interface planning and RL.”
>
> ## Q6: Minor comments
> Thank you for pointing these out. We fixed them in the revised version.

---

> ### Author Response · Authors · 2021-08-15
> **Thank you for your updated review! We made additional updates to the paper.**
>
> Thank you for your additional constructive feedback, and for updating the score! We have since updated the paper again, also addressing your response to our initial update. Therefore, please allow us to answer below:
>
> ## Comment 1: More directly related learning+planning algorithms
> We agree. We followed this suggestion and added an additional learning+planning baseline method to the experiments.
>
> In the original version of the paper, one of the baselines we compared L2E with was plan execution with an inverse model (“Plan + IM” baseline). This baseline method, like L2E, combines planning and learning. We now updated the paper again to include an additional baseline that combines planning and learning as well. Here we use the planner to create a sequence of subgoals that can be navigated by a RL agent. This approach is inspired by PRM-RL [Faust et al. 2017]; we will refer to it as “Planned Subgoals + RL”. In the pushing environment included in the original version of the paper, we find that this baseline outperforms all other baselines that we initially included, but still shows lower performance than both versions of L2E. In the obstacle environment, we find that “Planned Subgoals + RL” performs significantly worse than the L2E agents, as well as significantly worse than the “Plan + IM” baseline.
>
> More details are presented in an anonymized way here: https://github.com/paper6317authors/paper6317. We added a subsection to section “5.3 Baselines”, in which we introduce the “Planned Subgoals + RL” baseline in detail. We updated the figures and results description (section 5.4). We also updated the discussion (section 6) to now discuss this baseline as well.
>
> ## Comment 2: Custom made environments
> We agree that it would be desirable to compare L2E to environments that are not custom made. We present L2E as a method to learn to execute plans in hard-exploration physical manipulation tasks, for which a crude planner is available (or can be created). The widely used openAI gym benchmark contains 4 experiments in this category (https://gym.openai.com/envs/#robotics). Apart from a very simple pick-and-place task, these tasks are all pushing related.
>
> The pushing task we presented in the original version of the paper is almost equivalent to the openAI gym FetchPush-v1 task (same state and action spaces, same dynamics). Since both are alike, we decided to choose our version because it is open source, and will make it available on our github.
>
> The custom-made obstacle environment was chosen to represent robotic manipulation tasks with segmented state spaces. We intended to cover a more representative variety of challenges by using this environment, allowing for a better understanding of the limitations of L2E. We chose this environment over an environment from the openAI gym benchmark because we would argue that in contrast, the openAI gym scenarios are relatively similar to each other and to what we included initially, not allowing for an investigation of these limitations.
>
> Having said that, we agree that these decisions should be explained in more detail, and therefore added the following text in the experiments section in line 198:
>
> “We evaluate the L2E agent against several baselines using two robotic manipulation tasks, namely a pushing task and an obstacle avoidance task. These two environments are chosen to compare different approaches on a wide variety of challenges. While the pushing task can be seen as an open-source version of the opanAI gym FetchPush-v1 task [Brockman et al. 2016], the obstacle task was chosen to represent robotic manipulation tasks with segmented state spaces. This allows us to discuss limitations of exploration in such environments as well.”
>
> ## Comment 3: Plan-conditioned vs. goal-conditioned
> We agree that there are many approaches to create multi-modal policies. However, we would argue that the important distinction between L2E and other approaches for multi-modal policies is that the L2E policy is conditioned on plans as additional latent variable. If e.g. the environment is changed, it suffices to update the planner’s model of the environment so that it creates plans that are feasible again. These can then be directly fed into the policy without retraining. We agree that this point was probably not clear from our initial response. We therefore changed the added text in the discussion (see also Q4):
>
> “As shown in the obstacle environment, an important difference between goal-conditioned policies and plan-conditioned policies is that the latter can learn to execute multiple plans that lead from the same initial state to the same goal. Thus, the policy learns multiple strategies to achieve the same outcome. In principle, this could allow it to adapt to changed scenarios where some of these strategies become infeasible. If e.g. the environment changes, it suffices to update the planner’s model of the environment so that it creates plans that are feasible again. These can then be directly fed into the policy without retraining.”

---

### Official Review · Reviewer_TpKX · 2021-07-02

**Rating:** 6
**Confidence:** 4

**Summary:**

This paper presents a method for combining plan-conditioned policies with reward-shaping and a given approximate planner to a sparse-reward, seemingly challenging robotic manipulation task. However, the paper is lacking in experiments, which is the main reason I am voting for rejection.

**Limitations And Societal Impact:**

L2E limitations are actually not discussed, despite a reference to the discussion in the checklist.


**Main Review:**

## Paper Strengths
**Intuitive Method**
This paper demonstrates that a plan-conditioned policy combined with an FV-RS reward shaping function is able to allow for better performance on goal-conditioned, sparse reward tasks. The method is novel, although the contribution isn’t too surprising.

**Hyperparameter Tuning**
The authors perform a proper hyperparameter tuning scheme for HER when comparing against the baseline.

**Performance Improvement**
The method presents a modest performance improvement over the baselines.

## Paper Weaknesses
**One environment, one task**
The authors should evaluate on at least 2, preferably 3+ environments/tasks to truly demonstrate the advantage of their method. This is one large reason I am currently not voting for acceptance.

**Method Clarity**
Question: Are plans embedded or given in full to the plan-conditioned policy?

**Ablation Studies**
How does planner quality affect L2E? How do plan lengths affect L2E? Can you show ablations here?

**FV-RS Introduction**
The $F_{FV}$ reward is discussed but not given an intuitive explanation. This section, and the method in general, would be more clear if all 3 terms were explained intuitively in Eq.3 and their effects on policy behavior were discussed explicitly.

**Missing Related Works**
Combining model-free and model-based RL is presented as a contribution, however there are works that have done this before (see for example, “When to Trust Your Model: Model-Based Policy Optimization by Janner et al, and “Temporal Difference Models” by Pong et al). The authors should add citations to these works and contrast L2E with them.


**Time Spent Reviewing:**

4

---

> ### Author Response · Authors · 2021-08-10
> **Thank you for your review!**
>
> ## Q1: One environment, one task
> We agree that more experiments would support the paper. We therefore added an additional task in the revised version. While the pushing environment in the original version of the paper is an example with an “unsegmented” state space, we now also consider a scenario where (1) an obstacle needs to be evaded and (2) stochasticity is increased significantly. Obstacle avoidance is a notoriously hard problem in robotics. The obstacle "segments" the state space, posing a significant challenge to exploration. We find that L2E shows improved performance compared to the HER baselines and the pure planning baseline. While currently the L2E runs are not converged yet, preliminary results show that the performance of L2E is already comparable to the performance of planning with a learned inverse model.
>
> The experiments are described and preliminary results are presented in an anonymized way here: https://github.com/paper6317authors/paper6317 Due to restricted space, we added the experimental details for these new environments in the appendix, and reference them in the experimental section. We updated the discussion (section 6) to now discuss the new experiments as well.
>
> ## Q2: Method Clarity: Plan embedding
> Plans are embedded (see added text below for details). We agree that the discussion of this was too cursory (mentioned in section 5.2.). In the revised version, we clarified this point in more detail by adding a subsection before 5.2. (see below for the full text; square brackets are omitted in the paper).
>
> Furthermore, we added additional experiments, comparing analytical encoding as used originally to (1) using no encoding and (2) learning the encoding with a variational autoencoder (VAE) using a mean squared error reconstruction loss on the plan. See below for full results. We find that, while the analytical embedding performs significantly better than learning it using a VAE, a learned embedding is still advantageous compared to using no embedding at all (giving full plans to the policy).
>
> The experiments are described and preliminary results are presented in an anonymized way here: https://github.com/paper6317authors/paper6317. Due to restricted space, we added the experimental details for these new experiments in the appendix, and reference them in the experimental section.
>
> We added the following text before 5.2.:
>
> “5.2. Plan Encoding
>
> The plans p are embedded before they are provided to the policy. A plan encoding is an injective function \phi: \mathbb{P} \rightarrow \mathbb{C} from the set of plans \mathbb{P} to a latent space \mathbb{C}. If \mathbb{P} is a manifold in some high-dimensional space, the dimensionality of the latent space must therefore be at least as high as the dimensionality of the manifold. Since \mathbb{P} is task-dependent, the encoding will be task-dependent as well. For the pushing environment without obstacle [included in the original version], the plans are on a 4D manifold (since they only depend on the initial and final 2D box positions). For the obstacle task [added in the revised version], plans are on a 6D-manifold (since they depend on one intermediate box position as well).
>
> In the experiments discussed in the present work, we encode plans analytically using box positions as described above. We present experimental comparisons on either learning the encoding or not using any encoding at all in section A.3.“
>
> ## Q3: Ablation on planner quality
> We added additional experiments, comparing different plan lengths, in the revised version. Shorter plans (less dense plans) result in a less informative reward signal for the RL agent. We find that while longer (more dense) plans increase sample efficiency particularly in the beginning, L2E is largely invariant to plan length.
>
> The experiments are described and preliminary results are presented in an anonymized way here: https://github.com/paper6317authors/paper6317 Due to restricted space, we added the experimental details for these new experiments in the appendix, and reference them in the experimental section.
>
> ## Q4: FV-RS Introduction
> In the revised version, we appended section 3 with explicit/intuitive explanations on the effect of all three different terms used in eq. (3). We inserted the following text after line 128:
>
> “The first term in eq. (3) ensures that the assigned shaping reward is always smaller than the maximum environment reward (at most 1/2), and that if the binary environment reward is 1, no shaping reward is assigned. The second term rewards the agent for advancing towards the goal along the plan, and the third term rewards the agent for staying close to the plan.”
>
> ## Q5: Missing Related Works
> Thank you for mentioning this point. It was not our intention to present combining model-based and model-free RL as a contribution, since, as you pointed out, there is a large body of existing work on it. We believe that maybe lines 31-32 could have been ambiguous, and we therefore updated the last sentence to:
>
> “In the present work we seek to add an additional perspective to the open question of how this can be achieved best.” We hope that this clarifies our statement of contribution.
>
> Second, we agree that L2E, being at the intersection of planning and learning, is related in spirit to other work at the intersection of model-based and model-free methods as well. The difference of L2E to existing work is how the model information is included (via a planning step and FV-RS) and how this is then used to train a plan-conditioned policy. We updated the related work section to dedicate more discussion to this line of research in the revised version, including (among others) the references you have mentioned.
>
> Specifically, we added the following text to the newly created related work section “2.4. Combining Learning and Planning”:
>
> “[...] While the aforementioned work relies on planning to integrate models with model-free RL, the idea of integrating model-based and model-free RL also has been studied independently of planning ([Pong et al. 2018], [Janner et al. 2019]). In contrast, in L2E the model is translated by a planner into long-horizon plans.”
>
> ## Q6: Limitations
> We initially included the discussion of limitations in section 6 (Discussion and Conclusion), where we e.g. discussed the applicability to sparse-reward settings and the use of analytical plan encodings. However, we agree that limitations should be discussed more prominently and in more detail. We therefore split up section 6 into a conclusion and a section dedicated to limitations. We also added several aspects that came up during the review process, including (1) a discussion of limitations related to the use of suboptimal plans with FV-RS and (2) a discussion of limitations related to very large plan spaces. Specifically, we inserted the following text:
>
> “The use of FV-RS biases the RL agent towards following the plan. While it was shown in the experiments that the RL agent can learn to deviate from the plan, plans that are globally misleading can act as a distraction to the agent. In the present work, it is assumed that plans can be used to guide the agent during learning, increasing sample efficiency. Independently of the specific method used to achieve this, misleading plans will always break this assumption.
>
> As can be seen when comparing the pushing environment without obstacle to the obstacle environment, the amount of data needed for learning a plan-conditioned policy depends on the size of plan spaces that are learned. For very large plan spaces, more data will be needed to master the task. Still, including planner information into the learning process makes a decisive difference, as demonstrated by the relative performance of L2E and HER in the obstacle example.”

---

> > ### Comment · Reviewer_TpKX · 2021-08-13
> > **Addressed my concerns**
> >
> > Thank you for addressing my concerns and writing the specific text changes that will be made to address the clarity/related works issues I had with the paper.
> >
> > The new experiment is meaningful, although I was still hoping that the environment was going to be more distinct from the currently existing environment. However, given the brief time the authors had for performing the experiment, this is understandable.
> >
> > In response to the author's changes and new experiments, I will be raising my score to a 6.

---

> > > ### Author Response · Authors · 2021-08-15
> > > **Thank you for your answer!**
> > >
> > > Thank you for your additional constructive feedback, and for updating the score! We have since updated the paper again, also addressing your remark on the additional environment we added. Please let us quickly elaborate below:
> > >
> > > ## Comment 1: New environment
> > >
> > > Thank you for agreeing that the new experiment is meaningful. Just to briefly explain our choice: We present L2E as a method to learn to execute plans in hard-exploration physical manipulation tasks for which a crude planner is available (or can be created). We believe that obstacle avoidance is a particularly difficult example of such tasks, since the segmentation of the state space poses a significant challenge to exploration. We updated the paper with more details on the motivation for our choice of experiments, adding the following text in the experiments section in line 198:
> > >
> > > “We evaluate the L2E agent against several baselines using two robotic manipulation tasks, namely a pushing task and an obstacle avoidance task. These two environments are chosen to compare different approaches on a wide variety of challenges. While the pushing task can be seen as an open-source version of the opanAI gym FetchPush-v1 task [Brockman et al. 2016], the obstacle task was chosen to represent robotic manipulation tasks with segmented state spaces. This allows us to discuss limitations of exploration in such environments as well.”

---

### Official Review · Reviewer_PkED · 2021-07-17

**Rating:** 6
**Confidence:** 3

**Summary:**

The paper proposes Learning to Execute (L2E), which can make use of a pre-designed planner in reinforcement learning (RL) in robotics. The authors extend the idea of final-volume-preserving reward shaping (FV-RS) and formulate plan-conditioned MDPs for designing L2E. The method was evaluated through a simulation experiment.

**Limitations And Societal Impact:**

The paper is motivated by the real-world robot application of RL.  (At least, the authors insisted so.) Unfortunately, the approach and discussion do not seem to grasp the nature of the real world and actual situations in robotics. That seems to be noted to start the discussion of the sample efficiency. A critical issue of this paper is the unclear definition of "model-based planner." It is quite unclear to me. What kind of controller system and settings in robotics do the authors assume?  As far as I read through the paper, it is some idealistic Mathematical element.
Specifically, I am not sure what kind of "model-based planner" can satisfy the condition (5). The goal of a planner (or a controller) in state space is not always available in robotics. At least, the uncertainty of the environment and some clear and realistic assumptions should be considered.   Of course, in a simulation environment and a simple scenario like an experiment in this paper (which we may call a "toy problem"), we can find a setting satisfying the condition. If "model-based planner" in robotics does not satisfy the condition (5), the approach is not valid for solved the problem raised at the beginning of this paper.
Also, the term "model-based" is also used in "model-based RL," and policy is also called planner in some context. From these views, the authors should clarify the definition of "model-based planner" in this paper and show examples of them in robotics.

The comparison is limited. The method is about the integration of a pre-designed policy and RL via sub-reward. In comparison, I recommend the authors to compare the proposed method using other types of sub-reward approach, e.g., [Andrew 1999].

Also, this study is closely related to the integration of imitation learning (IL) (i.e., learning from demonstration or expert information) and reinforcement learning. For example, the following paper integrates IL and RL via sub-reward based on control as an inference framework. The relationship is worth mentioning.

- Akira Kinose, Tadahiro Taniguchi, Integration of imitation learning using GAIL and reinforcement learning using task-achievement rewards via probabilistic graphical model, Advanced Robotics, 34(16), pp.1055-1067, 2020. DOI: 10.1080/01691864.2020.1778521




**Main Review:**

The paper has certain originality regarding the idea of introducing FV-RS, especially from the theoretical viewpoint.
However, the validity of this method was not demonstrated sufficiently. Comparison with more pre-existing methods is expected.

The quality of this paper is generally reasonable.
The clarity of this paper is also reasonable.

However, the contribution is not significant. It is incremental because it is a straightforward application of FV-RS. For that application, some assumptions, which seem unrealistic in the real-world robotics environment, are introduced.  This point relates to the definition of "model-based planner" in this paper, which will be discussed in this review's "limitation" part.
At least, the significance is not demonstrated sufficiently through the experiment.

**Time Spent Reviewing:**

6 hours

---

> ### Author Response · Authors · 2021-08-10
> **Thank you for your review!**
>
> ## Q1: Application of FV-RS
> We agree that we have not clearly emphasized why L2E is in our mind not merely an application of FV-RS: The core idea of this paper is to learn policies that are conditioned on plans from a planning algorithm instead of learning goal-conditioned policies. This is complemented by a study of plan sampling strategies. Such a plan-conditioned policy could be applied to any plan-conditioned MDP, and FV-RS is not necessarily an integral part of this proposal. In the experiments, we use FV-RS as a tool to (a) construct goal-conditioned MDPs from plan-conditioned MDPs for a better comparison to goal-conditioned approaches and (b) to speed up learning. We agree that this point was unclear; we therefore updated lines 39-41 in the statement of contribution in the following way:
>
> “We introduce L2E, which uses RL to learn plan-conditioned policies to execute approximate plans from a model-based planner in a plan-conditioned MDP. We describe formally how FV-RS can be used as a tool to construct such plan-conditioned MDPs from goal-conditioned MDPs.”
>
> ## Q2: Definition of "model-based planner" and how it can be used in the real world
> Thank you. In our formulation, a model-based planner is any module that outputs a plan to get from an initial state to a final state that satisfies the goal g. We call it model-based since it must have some (albeit very basic) prior knowledge of the environment to be able to do so. However, its knowledge of the environment does not have to be perfect; the resulting plan is only used as a guidance for the RL algorithm. Examples of such model-based planners include trajectory optimization (e.g. [Toussaint 2015]) and probabilistic roadmaps [Kavraki et al. 1996], and L2E is agnostic to the exact approach chosen.
>
> We would argue that it is not uncommon in real-world robotics situations that an approximate model is available, as this was studied in various contexts such as indoor and aerial robots [Faust et al. 2014], visual manipulation [Jeong et al. 2020], or humanoid walking [Mordatch et al. 2015]. While simply executing on the resulting approximate plan is insufficient due to inaccuracies of the underlying model, L2E can be used to robustly learn to execute such plans. We updated the introduction of the paper to discuss our notion of "model-based planner" more clearly. We replaced lines 22-26 with the following:
>
> “On the other hand, for many robotics scenarios, there is a rough model of the environment available. This can be exploited, e.g., using model-based planning approaches (Mordatch et al., 2012; Kuindersma et al., 2016; Toussaint et al., 2018). Such model-based planners potentially offer a more data-efficient way to reason about an agent’s interaction with the world. Model-based planners have been used in many areas of robotics, such as for indoor and aerial robots (Faust et al. 2018), visual manipulation (Jeong et al. 2020), or humanoid walking (Mordatch et al. 2015). Still, if the model does not account for stochasticity or contains systematic errors, directly following the resulting plan will not be successful.”
>
> ## Q3: Condition (5) is unrealistic
> Thank you. Here we kindly disagree. We would argue that condition (5) can always be satisfied without loss of generality (see explanation below). Basically, we stated condition (5) only to simplify notation. That being said, we agree that it has probably been formulated in an unclear way, and we updated this in the revised version.
>
> Explanation: Condition (5) simply states that the plans are such that the goal can be recovered from it. In most cases it will simply be the final state in a sequence of planned states. However, the goal can be much more general and does not have to be in state space. Even if a planner should not satisfy condition (5), its output can be simply “concatenated” with the original goal, which results in a planner that satisfies condition (5) (see inserted text below for details). Therefore, we argue that condition (5) is not restrictive, but on the contrary can be assumed to be true without loss of generality. The reason we stated it was to simplify notation, but we agree that this has probably complicated things too much. Therefore, we updated the revised version by replacing lines 150 - 152 (including eq. (5)) with the following:
>
> “We call g = f(p) the goal for which the plan p was created. If a planner \Omega should be such that g can not be recovered from the resulting plan p~\Omega(.|s,g), we can always construct a new \tilde{\Omega} such that \tilde{p} = [p, g]. As now g can be recovered from \tilde{p} deterministically, we can assume that f always exists without loss of generality.”
>
> ## Q4: Clarification of model-based planners vs. model-based RL
> We agree that our usage of "model-based planner" should be clarified.  As detailed in our answer to Q2, we updated the introduction of the paper to discuss our notion of "model-based planner" more clearly.
>
> ## Q5: Comparison with more pre-existing methods: Compare to potential-based reward shaping [Andrew Ng et al. 1999]
> We agree that the comparison of FV-RS to potential-based reward shaping [Andrew Ng et al. 1999] is relevant for L2E. However, a thorough comparison of FV-RS to [Andrew Ng et al. 1999] was already performed in the original FV-RS paper. There it was found in several plan-based settings that FV-RS significantly and robustly outperforms [Andrew Ng et al. 1999]. We would argue that repeating such a comparison in the present paper would not be a novel contribution. We agree however that the results of this comparison are still relevant for L2E, and are worth mentioning. In the revised version, we therefore included the discussion of the comparison to [Andrew Ng et al. 1999] in section 6:
>
> “[...]. In the present paper, FV-RS was used to increase the sample efficiency of L2E. There are alternative schemes of reward shaping such as potential-based reward shaping (PB-RS) [Ng et al. 1999]. These could also be used to increase the sample efficiency of learning a plan-conditioned policy in any given plan-conditioned MDP. We chose FV-RS since in the original paper [Schubert et al. 2021], it was demonstrated that FV-RS leads to significantly higher sample efficiency than PB-RS. [...]”
>
> ## Q6: Related work
> Thank you for pointing this out. We agree that this is interesting related work. We also agree that the integration of RL and Learning from Demonstration is related, and discussed several references in section 2.3 of the original version. We split up section 2.3 into “2.3 Learning from Demonstration” and “2.4. Combining Learning and Planning” in the revised version. We included the reference in the related work section 2.3 in the following way:
>
> “[Kinose & Taniguchi 2020] integrate RL and demonstrations using generative adversarial imitation learning by interpreting the discriminator loss as another optimality signal in multi-objective RL. [...]While LfD approaches are related to L2E in the sense that external information is used to increase RL efficiency, it is in contrast assumed in L2E that this external information is provided by a planner.”

---

> > ### Comment · Reviewer_PkED · 2021-08-25
> > **Updated**
> >
> > Thank you very much for your clear responses.
> > I noticed some misunderstandings especially about (5).
> > I also welcome many clarifications you made, and appreciate your effort to conduct an additional experiment.
> >
> > Based on the update, I changed my score from 5 to 6.

---

### Official Review · Reviewer_4YB3 · 2021-07-18

**Rating:** 7
**Confidence:** 5

**Summary:**

This paper addresses the problem of sample complexity in reinforcement learning. The approach is to drive down the sample complexity using reward shaping, where the reward shaping function is given by a planner. The paper shows how to use final-volume-preserving
reward shaping (FV-RS) to convert a plan into a reward shaping function. The approach is evaluated on a simple pushing domain against planning in two forms, and Hindsight Experience Replay. The L2E approach proposed here outperforms the comparison algorithms.


**Limitations And Societal Impact:**

I appreciated that the author checklist pointed at the specific section where they were attempting to discuss limitations. Of the papers that I reviewed, this paper probably provided the best analysis of the limitations of the work. That being said, the discussion of limitations was quite cursory. There is room for considerably more analysis of the limitations of this work, especially in the experimental stage. It seems clear that the biggest limitation of the work is the loss in optimality induced by FV-RS. I expect this loss in optimality will create very negative consequences when the optimal plan and optimal policy are wildly divergent.

For instance, imagine a world with two routes to the goal, where one route is highly stochastic with very high expected cost, but is much shorter, e.g., a short cut along a cliff’s edge. The other route is much longer is far less stochastic (or just has no negative outcomes), so much lower expected cost (a route away from the cliff’s edge). if the planner assumes a deterministic world, it may incorrectly assume that the cost of the path along the cliff’s edge is much lower than the cost of the longer route, and very strongly bias the learner in the wrong direction.

This is not to say that this paper is wrong or incorrect in some way, but that obvious limitation does have to be addressed.

The author checklist indicated that the authors did not discuss any potential negative societal impacts of the work.


**Main Review:**

Overall, I like the idea of this paper and would argue for its acceptance. The technical ideas are strong, and the paper is (mostly) well-written. The primary limitation of this work is the experimental evaluation, especially in an increasingly crowded field of prior work combining planning and learning.  It is not clear to me that the most relevant baseline is HER. There are a number of more relevant works, such as PRM-RL (Faust et al, 2014) and Hoel et al (2020), both of which explicitly combine planning and learning for exactly the same purpose as L2E, and there are other references as well. I think the real novelty is the use of FV-RS as the shaping function, but the other papers in this line of research need to be considered.

I would recommend substantially strengthening the experimental evaluation. The single domain of box pushing is not particularly compelling, and does not show off the strength of incorporating a planner in the reward shaping function — what would be better would be problems with a much longer horizon and places where a significant deviation from a greedy strategy would be important, but with stochasticity to be avoided. (BTW: the plan shown in figure 1 was very hard to interpret. I now realize that the reason the end effector is changing height is because it is going over the box to change sides to push, but it took quite a while to figure that out. Some more explanatory text would help considerably. Maybe also draw the box at different along the trajectory with a significant alpha channel value.) Figure 1b was not a particularly effective way of communicating the results. It was not clear to me if these are the average and variances of the 10 agents on one problem, or the average and variances of the 10 agents times 30 random start/goal states per agent.

Even for these experiments, I would recommend running L2E and HER longer — it is not clear to me that L2E $ S^{bias} _{10,1000}$ has converged, especially when looking at 1d. I assume the loss in performance for 10 plans from 1000 samples is because it has experienced fewer transitions. And if indeed L2E $ S^{bias} _{10,1000}$ has converged, I would want to know why it is not exploring further.

Equation 3 is very important, and it is confusing and feels somewhat arbitrary — the paper relies too much on explication from the FV-RS paper, including replicating notation without explanation. The fact that f(p) is the final state of the plan is never stated, although the fact that f(p) is the goal on the following page allows the reader work backwards. $p_{k(s)}$ should be explained, as well as the fact that the subscript is indexing the plan state. Even with the the FV-RS paper in hand, the justification for the $k(s)+1/L$ term is unclear to me. I would encourage the authors to add explanatory text for the terms in the FV-RS reward shaping equation.

**Time Spent Reviewing:**

1.5 hours

---

> ### Author Response · Authors · 2021-08-10
> **Thank you for your review!**
>
> ## Q1: Related work on combining planning and learning
> Thank you for this remark. We agree that L2E, being at the intersection of planning and learning, is related to other work at this intersection as well. We would like to point out that we therefore, in addition to HER, also compare L2E to learned inverse model plan execution [Christiano 2016] in the experiments section; this is a baseline method that combines planning and learning. Purely learning-based approaches do not run the risk of being biased by a wrong model, which is why we compared to HER as well. That being said, we agree that the line of work on combining planning and learning is worth giving more emphasis in the related work section. We therefore split up section 2.3. into “2.3. Learning from Demonstration” and “2.4. Combining Learning with Planning” in the revised version. We couldn't find the publication by Hoel et al. 2020, and we'd be glad to include it if you provide us a pointer for the mentioned paper. Apart from that, we included (among others) the references you have mentioned. We inserted the following text after line 87:
>
> “2.4. Combining Learning with Planning
>
> Similarly to demonstrations, external plans can be exploited to facilitate learning. [Faust et al, 2018] connect short-range navigation policies into complex navigation tasks using probabilistic roadmaps. In contrast, L2E learns a monolithic plan-conditioned policy for the entire task, using planning only to speed up the process. [Sekar et al. 2020] use planning in a learned model to optimize for expected future novelty. In contrast, L2E encourages the agent to stay close to the planned behavior. [Zhang et al, 2016] use model-predictive control to generate control policies that are then used to regularize the RL agent. In L2E, no control policy is created, and a reward signal is extracted directly from the plan. In Guided Policy Search [Levine & Koltun 2013], differential dynamic programming is used to create informative guiding distributions for policy search from a transition model. These distributions are used to directly regularize the policy in a supervised fashion, while L2E makes use of FV-RS as a mechanism to interface planning and RL.”
>
> ## Q2: Strengthen experimental evaluation
> We agree that more experiments would support the paper. We therefore added an additional task in the revised version. While the pushing environment in the original version of the paper is an example with an “unsegmented” state space in which greedy behavior is successful most of the time, we now also consider a scenario where (1) an obstacle needs to be evaded and (2) stochasticity is increased significantly. Obstacle avoidance is a notoriously hard problem in robotics. The obstacle "segments" the state space, posing a significant challenge to exploration. We find that L2E shows considerably improved performance compared to the HER baselines and the pure planning baseline. While currently the L2E runs are not converged yet, preliminary results show that the performance of L2E is at least comparable to the performance of planning with a learned inverse model.
>
> The experiments are described and preliminary results are presented in an anonymized way here: https://github.com/paper6317authors/paper6317. Due to restricted space, we added the experimental details for these new environments in the appendix, and reference them in the experiments (section 5). We updated the discussion (section 6) to now discuss the new experiments as well.
>
> ## Q3: Figure 1a is hard to read
> Thank you, we (1) added the sentence “The end effector is planned to change height to push the box from the other side.” to the caption of figure 1a, and (2) will add transparent boxes along the trajectory shown in figure 1a, as suggested.
>
> ## Q4: Figure 1b is not particularly effective
> To first answer your question: the latter is the case; figure 1b shows averages over 10 agents times 30 random start/goal configurations, and the confidence interval of these averages. We agree that this was hard to see from the figures (only indicated by the sum indices in the y-axis caption), thus we updated the y-axis caption to “Success Rate Averaged over Agents and Initial Configurations”, dropping the sum formula in the caption. We also updated the second sentence of the caption to “The lines represent averages over N independently trained agents times M test rollouts per agent and time step, the colored areas indicate the confidence intervals [...].”
>
> ## Q5: Run L2E experiments for longer
> Agreed. We will include figures with around 10 times more steps (60 Million) in the revised version, to allow for a more direct comparison between the two versions of our method. These runs are not finished yet, but preliminary results indicate that both L2E 100-1000 and L2E 10-1000 indeed improve further.
>
> ## Q6: Notation in eq. (3) unclear
> Thank you. We fixed this and now introduce f(p) and the meaning of the indexing p_k(s) directly after eq. (3) in the revised version. We also added the following explanations on the effect of all three different terms used in eq. (3) after line 128:
>
> “The first term in eq. (3) ensures that the assigned shaping reward is always smaller than the maximum environment reward (at most 1/2), and that if the binary environment reward is 1, no shaping reward is assigned. The second term rewards the agent for advancing towards the goal along the plan, and the third term rewards the agent for staying close to the plan.”
>
> ## Q7: Discussion of Limitations
> We agree that the paper could be strengthened by a deeper discussion of limitations at the experimental stage. We split up section 6 into a conclusion and a section dedicated to limitations, where we discuss, among others, data requirements of L2E in large plan spaces. This is based on a comparison of the environment in the original version of the paper to the environment we added now. We also agree that the loss in optimality of FV-RS is a limitation that L2E inherits: L2E will learn to correct inaccuracies of the plan (as seen in the experiments), but if a plan is “globally wrong” in the sense you describe, the shaping reward can act as a distraction for the learned policy. We appended the the limitations section with the following discussion of this limitation:
>
> “The use of FV-RS biases the RL agent towards following the plan. While it was shown in the experiments that the RL agent can learn to deviate from the plan, plans that are globally misleading can act as a distraction to the agent. In the present work, it is assumed that plans can be used to guide the agent during learning, increasing sample efficiency. Independently of the specific method used to achieve this, misleading plans will always break this assumption.”

---

> > ### Author Response · Authors · 2021-08-15
> > **We made additional updates to the paper (added baseline) which relate to Q1 of your review.**
> >
> > ## Q1: Related work on combining planning and learning
> >
> > Thank you again for this remark. Following your suggestion, we added an additional baseline method combining learning and planning to the experiments. This is in addition to the updates to our related work section, which we mentioned in our initial answer to Q1.
> >
> > In the original version of the paper, one of the baselines we compared L2E with was plan execution with an inverse model (“Plan + IM” baseline). This baseline method, like L2E, combines planning and learning. We now updated the paper again to include an additional baseline that combines planning and learning as well. Here we use the planner to create a sequence of subgoals that can be navigated by a RL agent. This approach is inspired by PRM-RL [Faust et al. 2017]; we will refer to it as “Planned Subgoals + RL”. In the pushing environment included in the original version of the paper, we find that this baseline outperforms all other baselines that we initially included, but still shows lower performance than both versions of L2E. In the obstacle environment, we find that “Planned Subgoals + RL” performs significantly worse than the L2E agents, as well as significantly worse than the “Plan + IM” baseline.
> >
> > More details are presented in an anonymized way here: https://github.com/paper6317authors/paper6317. We added a subsection to section “5.3 Baselines”, in which we introduce the “Planned Subgoals + RL” baseline in detail. We updated the figures and results description (section 5.4). We also updated the discussion (section 6) to now discuss this baseline as well.

---

> > ### Comment · Reviewer_4YB3 · 2021-09-11
> > **Thanks**
> >
> > Thanks for the detailed response. The additional experiments are helpful.

---

### Author Response · Authors · 2021-08-10
**Authors Initial Response - Overview**

We would like to thank the reviewers for their constructive and thorough feedback. We tried to address their concerns and revised the paper accordingly, and will also answer directly to their reviews. We are happy to further engage in the discussion during the rolling review phase. The following is an overview of all updates made:

1. We performed additional experiments, and included the results in the revised version. We summarize the results and discuss them in the direct answers below. Detailed plots are presented in an anonymized way here: https://github.com/paper6317authors/paper6317
Since some runs are not fully finished yet, we will continue to update this regularly.
    - We compare L2E to the baseline methods in an additional environment containing an obstacle to be avoided. We find that L2E significantly outperforms the HER baseline. Additionally, while our runs are not yet fully converged at this point, preliminary results show that the performance of L2E is already comparable to the performance of planning with a learned inverse model. [Answer to 4YB3, TpKX, CAJo]
    - We compare the encoding function we used in the original version to (1) using an encoding learned with a variational autoencoder (VAE) and (2) using no encoding at all. We find that, while the analytical embedding performs better than learning it using a VAE, a learned embedding is still advantageous compared to using no embedding at all. [Answer to TpKX, CAJo]
    - We compare the use of different plan lengths (densities) for L2E. We find that while longer (more dense) plans increase sample efficiency particularly in the beginning, L2E is largely invariant to plan length. [Answer to TpKX]
2. We split up section 6 into a discussion and a section dedicated to limitations. Also taking into account our new experimental results, we then added the discussion of several limitations:
    - “Distraction” of the L2E policy during training if the plans used are wrong in a global sense. [Answer to 4YB3, TpKX]
    - Required increase in data size when learning a plan-conditioned policy in very large plan spaces. [Answer to TpKX]
3. We split up section 2.3 into one section discussing related work on learning from demonstration and one section discussing related work on combining learning and planning. We added several references the reviewers suggested, and contrasted them with L2E. [Answer to 4YB3, PkED, TpKX, CAJo]
4. We added a justification of why we chose to use FV-RS over potential-based reward shaping to section 6. [Answer to PkED]
5. We clarified our use of the term “model-based planner” in the introduction, referencing real-world examples in robotics. [Answer to PkED]
6. We added an explicit explanation of the effects of the different terms in the FV-RS shaping function introduced in eq. (3). [Answer to 4YB3, TpKX]
7. We added a subsection before section 5.2 to describe the plan encoding used in the experiments in detail (see also 1b). [Answer to TpKX, CAJo]
8. We updated the first point of our statement of contribution to clarify the relation of L2E to FV-RS. [Answer to PkED]
9. We updated line 31-32 to clarify how L2E relates to existing work on combined model-based and model-free RL. [Answer to TpKX]
10. We appended section 6 with a general discussion of the advantages of learning a plan-conditioned policy over a goal-conditioned policy, and clarified a statement made in lines 272-274. [Answer to CAJo]
11. We updated our writing around eq. (5) to explain more clearly that condition (5) can be assumed without loss of generality. [Answer to PkED]
12. We started longer runs for the experiments in the original version of the paper in order to better compare different versions of our method. [Answer to 4YB3]
13. We incorporated various minor comments by the reviewers. [Answer to 4YB3, CAJo]
14. We moved experimental details as well as the discussion of hyperparameters for the HER baseline to the appendix to create space for the added experiments and discussion.

---

> ### Author Response · Authors · 2021-08-22
> **Updated experimental data to 1: L2E significantly outperforms baselines**
>
> Since the initial response, the calculations for the obstacle environment have progressed. While the previously reported preliminary results already indicated that L2E performs at least comparable to the best baseline in the obstacle environment, the updated results show that L2E significantly outperforms all baselines (including the added “Planned Subgoals + RL” baseline). More details are presented in an anonymized way here: https://github.com/paper6317authors/paper6317. [Answer to 4YB3, TpKX, CAJo]

---

### Author Response · Authors · 2021-08-15
**Authors Second Response - Overview**

We would like to thank the reviewers for their constructive responses to our initial answers and updates. We will also answer directly to their responses. We are happy to further engage in the discussion during the rolling review phase. We made the following additional updates to the paper:

1. We added an additional baseline method to the experiments. In the original version of the paper, one of the baselines we compared L2E with was plan execution with an inverse model (“Plan + IM”). This baseline method, like L2E, combines planning and learning. We now updated the paper again to include an additional baseline that combines planning and learning as well. Here we use the planner to create a sequence of subgoals that can be navigated by a RL agent. This approach is inspired by PRM-RL [Faust et al. 2017]; we will refer to it as “Planned Subgoals + RL”. In the pushing environment included in the original version of the paper, we find that this baseline outperforms all other baselines that we initially included, but still shows lower performance than both versions of L2E. In the obstacle environment, we find that “Planned Subgoals + RL” performs significantly worse than the L2E agents, as well as significantly worse than the “Plan + IM” baseline. More details are presented in an anonymized way here: https://github.com/paper6317authors/paper6317. [Answer to 4YB3, CAJo (updated review after rebuttal)]
2. We motivated our choice of environments in more detail at the beginning of the experiments section. [Answer to CAJo (updated review after rebuttal), TpKX (response to rebuttal)]
3. We clarified the discussion of the advantages of learning a plan-conditioned policy over a goal-conditioned policy in section 6. [Answer to CAJo (updated review after rebuttal)]

---

### Decision · Program_Chairs · 2021-09-27

**Decision:**

Accept (Poster)

**Comment:**

The paper combines learning and planning to increase data efficiency, motivated by robotics application scenarios. In more detail, the authors leverage the fact that in such scenarios, often, a coarse model is available, that allows for making a coarse, but probably suboptimal plan. The authors propose using this plan to provide shaping rewards to a RL agent that is then free to further optimize the policy. For this, they build on "plan based final volume preserving reward shaping" introduced by Schubert et al. The main contribution of the authors is to
- Allow policies that generalize over instances by conditioning on the plan
- Using this formulation to introduce plan-replay-strategies.

Initially, the reviews was mixed, but after the discussion phase all reviewers recommend acceptance. The main reasons being:
- Strong and original technical ideas
- Mostly well-written
- The experimental evaluation was initially somewhat limited, but during the discussion phase the authors have made these stronger by adding another environment and clarifying the baseline methods used.
- The methods limitations could be more explicitly discussed.

I'd like to add that I think that the authors could more explicitly discuss what is novel in their paper compared to the earlier Schubert paper. E.g., the FV-RS is directly introduced in a 'universal' formulation (3), making the transition to universal policies and reward shaping functions more smooth but also making this part of their contribution less explicit.